# Clinical Targeted Panel Sequencing Analysis in Clinical Evaluation of Children with Autism Spectrum Disorder in China

**DOI:** 10.3390/genes13061010

**Published:** 2022-06-02

**Authors:** Chunchun Hu, Linlin He, Huiping Li, Yanhua Ding, Kaifeng Zhang, Dongyun Li, Guoqing Zhu, Bingbing Wu, Xiu Xu, Qiong Xu

**Affiliations:** 1Department of Child Health Care, Children’s Hospital of Fudan University, Shanghai 201102, China; 14211240007@fudan.edu.cn (C.H.); lihuiping@fudan.edu.cn (H.L.); yeah_ding@aliyun.com (Y.D.); 2015gantong@sina.com (K.Z.); dongyunli@fudan.edu.cn (D.L.); xuxiu@fudan.edu.cn (X.X.); 2Pediatric Department, Suining Central Hospital, Suining 629000, China; anita0918@sina.com; 3Pediatric Department, Binzhou Peoples’ Hospital, Binzhou 256600, China; kad_zhuguoqing@163.com; 4Clinical Genetic Center, Children’s Hospital of Fudan University, Shanghai 201102, China; bingbingwu2010@163.com

**Keywords:** autism spectrum disorder, targeted panel sequencing, genetic variants, *MYT1L*, *SLC26A4*

## Abstract

Autism spectrum disorder (ASD) is an early-onset neurodevelopmental disorder in which genetics play a major role. Molecular diagnosis may lead to a more accurate prognosis, improved clinical management, and potential treatment of the condition. Both copy number variations (CNVs) and single nucleotide variations (SNVs) have been reported to contribute to the genetic etiology of ASD. The effectiveness and validity of clinical targeted panel sequencing (CTPS) designed to analyze both CNVs and SNVs can be evaluated in different ASD cohorts. CTPS was performed on 573 patients with the diagnosis of ASD. Medical records of positive CTPS cases were further reviewed and analyzed. Additional medical examinations were performed for a group of selective cases. Positive molecular findings were confirmed by orthogonal methods. The overall positive rate was 19.16% (109/569) in our cohort. About 13.89% (79/569) and 4.40% (25/569) of cases had SNVs only and CNVs only findings, respectively, while 0.9% (5/569) of cases had both SNV and CNV findings. For cases with SNVs findings, the *SHANK3* gene has the greatest number of reportable variants, followed by gene *MYT1L*. Patients with *MYT1L* variants share common and specific clinical characteristics. We found a child with compound heterozygous *SLC26A4* variants had an enlarged vestibular aqueduct syndrome and autistic phenotype. Our results showed that CTPS is an effective molecular diagnostic tool for ASD. Thorough clinical and genetic evaluation of ASD can lead to more accurate diagnosis and better management of the condition.

## 1. Introduction

Autism spectrum disorder (ASD) is an early-onset neurodevelopmental disorder characterized by social communication deficits and repetitive sensory-motor behaviors that affect about 1% of children worldwide [1]. The reported sex ratio ranges from 2:1–5:1 (approximately 4:1) between males and females [2,3]. Genetic variants have been considered to be a major cause of pathogenesis. 64–93% ASD risk is heritable [4,5]. Studies of ASD siblings showed that 7–20% of subsequent children suffered ASD diagnosis after their elder brother/sister [6]. Risk also increased in families with a higher diagnostic rate of ASD.

Identifying genetic etiologies of ASD provides useful information for clinicians and families. Genetic testing keeps marching forward owing to advances in sequencing technologies. It is currently considered that ASD genetic variants are highly heterogeneous and individualized. Over 1000 genes have been reported to be associated with ASD [7]. Chromosomal aberrations, copy number variations (CNVs), and single nucleotide variations (SNVs) both play a role in the pathology of ASD and have led to progress in the understanding of the complex genetic background of the disease. Chromosomal microarray (CMA) analysis for CNV detection is recommended as the most appropriate initial test for the etiologic evaluation of ASD patients [8,9,10]. In recent years, whole-exome sequencing (WES) based on next-generation sequencing (NGS) technology has been applied for further etiological evaluation of patients without CMA findings. It allows for the identification of SNVs, including pathogenic substitutions, insertions, or deletions, which have been associated with ASD [8,11,12]. Whole-genome sequencing (WGS) has also added value as a diagnostic test for ASD [13]. Clinical heterogeneity is much more widely recognized in ASD children sharing core features [14]. Improving the skills of distinguishing key clinical features may increase the rate of recognition for patients harboring relevant genetic variants. In addition, children with certain syndromes may exhibit autism-like behaviors. Genetic testing can help diagnosis and evaluate prognosis.

Overall, finding a diagnostic etiology helps patients and families obtain more information about co-occurring medical problems and prognosis, acquire effective interventions and connect families to specific support groups. Therefore, a study using NGS containing CNVs and SNVs analysis for ASD cohort is ideal for clinical and research practice. In this study, we report the effectiveness and validity of clinical targeted panel sequencing (CTPS) comprising 2742 genes. Additionally, we report the yield and specific founding of CTPS cases and understand the correlations between genotype and phenotype of ASD.

## 2. Materials and Methods

### 2.1. Study Participants and Case Review

Patients receiving a diagnosis of ASD in the Department of Child Health Care, Children’s Hospital of Fudan University, were included consecutively from January 2019 to December 2020. The inclusion criteria were as follows: all the patients met the criteria of ASD diagnosed by developmental-behavioral pediatricians using the Diagnostic and Statistical Manual of Mental Disorders, fifth edition (DSM-V)) [15], medical records of positive CTPS results were reviewed. Additional medical examinations were performed for the selected cases. 

### 2.2. Clinical Targeted Panel Sequencing, Data Processing, and Variant Classification

Genomic DNA of every patient was extracted from peripheral blood samples in EDTA-coated Vacutainers according to standard procedure. CTPS was applied for enrolled ASD patients using the Agilent ClearSeq Inherited Disease panel kit (Santa Clara, CA, USA) for enrichment based on NGS [16,17]. The CTPS included 2742 genes. Sequencing was performed on an Illumina HiSeq X10 (Illumina, San Diego, CA, USA) with 150 bp pair-end sequencing. The average on-target sequencing depth was 200× for CTPS and average reads mapping rate was 99.8%. The fraction of the targeted region with at least 10× and 20× were 99.1% and 97.2%.

Details of the variant calling, filtration, and annotation can be found in our previously published papers [16]. Briefly, for SNV and small insertion/deletion calling, GATK best practice pipeline was applied, including sequence alignment to the hg19 reference genome by BWA (V.05.9-r16), sorting by Samtools (v.1.8) and the duplication removed by Picard (v.2.20.1), with default settings. Variants were annotated by VEP (v.104.2) [18] and ANNOVAR (v.2019-10-24) [19] with basic gene-based annotation (RefSeq, Ensembl), damaging prediction (SIFT, PolyPhen2), function annotation (OMIM), and pathogenicity annotation (ClinVar, HGMD). Then, variants were filtered according to the following criteria: (1) variants out of the capture region (exon region extended by 15 bp); (2) high allele frequency in public gnomAD databases; (3) zygosity not match; (4) variants from genes with AD inheritance model not inherited from healthy parents; (5) low-quality variants except reported pathogenic variants; (6) clinical phenotype matching by a computational phenotype filtering process. 

For CNV calling, two read-depth-based algorithms, CANOES and HMZDelFinder, were applied at the exon level and combined at the region-level. The PICNIC (Pipeline for clinical NGS-involved CNV detection) and AnnotSV (online version) were used for the following CNV filtration and annotation [20]. Variants were annotated with gene-based annotation (RefSeq), region-based annotation (DGV) and function annotation (OMIM). Then, variants were filtered, mainly considering the frequency of deletions or duplications in the internal samples, the region size and also the matching of clinical phenotypes. 

For the combination of SNV and CNV, an SCI strategy [16] was additionally applied to consider the complicated compound heterozygous condition. Finally, our internal automatic variant processing pipelines can leave an average of approximately a dozen genes (median 40) per sample for further manual review. 

We made the diagnosis considering both CNVs and SNVs based on the ACMG guidelines but with some adjustments, as published studies have described variant classification criteria [16,17]. The criteria of “diagnostic rate/positive rate” is the proportion of cases with positive findings, where we can find disease-causing SNVs or CNVs that can explain the patient’s phenotype with matched inheritance model. The “overall positive rate” is the proportion of cases finding both SNVs and CNVs. For the diagnostic SNVs, Sanger sequencing was used for variant confirmation, qPCR/MLPA was used for the diagnostic CNVs, and Mutation Surveyor software (SoftGenetics version5.0) was used to analyze the data for both patient samples and their parents. 

## 3. Results

### 3.1. Demographics and Clinical Files of Patients

A total of 573 patients receiving a diagnosis of ASD were enrolled in the cohort following the inclusion criteria. The sex ratio was 4.03:1 (459 Males to 114 females). The mean age of these patients was 3.6 years old, from 16 months to 12.8 years. A diagnosis of ASD was made by qualified developmental-behavioral specialists using DSM-V [15] and Autism Diagnostic Observation Schedule, second edition (ADOS-2) [21]. Figure 1 showed the flow of study.

Seventy patients underwent extra *FMR1* testing in addition to CTPS, who had prominent clinical features especially the facial characteristics such as long face, prominent ears, and prominent jaw, and 4 out of 70 (5.71%) obtained positive results and were lately diagnosed with Fragile X syndrome. Analyzing SNVs and CNVs simultaneously, an overall diagnostic yield of 19.16% (109/569) was reached (Table 1 and Table 2). SNVs alone were detected in 13.89% of the cases (79/569), 25 patients with CNVs alone accounted for 4.40% (25/569) of the detection rate, and the remaining 0.88% (5/569) had both SNV and CNV findings (Figure 2A).

For SNVs, 62 patients had missense mutations, 10 patients had frameshift mutations, 1 patient had a nonframeshift mutation, 4 had splicing mutations, and 7 had more than one kind of variant. We found that 5 patients had *SHANK3* variants, which was the greatest number of reportable variants, leading to the diagnostic yield of 0.88% (5/569). Four children had *MYT1L* variants. *MECP2*, *DIP2B*, *DYRK1A*, *FOXP1*, and *PHIP* variants were found in 3 patients, respectively (Table 1). For CNVs, 18 were duplication variants (interestingly, we found that 6 patients had 15q11-13 duplications in 20% of CNV cases), and 12 were deletion variants (all were heterozygous deletions), which are shown in Table 2 (for gene impacted and frequency of CNVs detailed in Appendix A).

Of 109 patients who had molecular abnormalities, 84 were males and 25 were females. For diagnostic SNVs in males, a total of 67 cases were included, of which “pathogenic” SNVs accounted for 8.96% (6/67), “likely pathogenic” SNVs accounted for 22.39% (15/67), and the remaining 68.66% (46/67) were “variants of unknown significance (VUS)”. CNVs were found in 21 male patients (including 4 patients with SNVs), of which 8 patients were also validated by CMA. In 25 females with genetic abnormalities, SNVs were found in 17 females and CNVs could be found in 9 females (5 had CMA validation), with one having both CNV and SNV. Meanwhile, the percentage of pathogenic SNVs in females was 11.76% (2/17), “likely pathogenic” was 29.41% (5/17), and “VUS” was 58.82% (10/17). There were no significant differences between the proportions of SNVs in males and females (Fisher’s exact *p* = 0.63) (Figure 2B). Thirty-four cases also conducted parental tests (both father and mother) by Sanger. Fourteen SNVs were de novo (41.18%, 14/34), 17 were inherited (50.00%, 17/34), and 3 had variants that were both de novo and inherited (8.82%, 3/34), leaving 50 unavailable to be tested.

### 3.2. Representative Cases from CTPS

#### 3.2.1. Summarized Cases: Analysis of Clinical Characteristics Classify Genetic Problems 

Among ASD patients who had identified SNVs, *SHANK3* was the most common variant (*n* = 5). The second on the list was *MYT1L*, a myelin transcription factor, a transcription factor enabling fibroblast-to-neuron conversions. The inheritance of Cases 19/21/25/26 was de novo (Figure 3A). Tracing the developmental milestones, they all had obvious delayed motor developments. The independent walking for 4 children ranged from 20 to 30 months. Additionally, all children had significant language delay, most of them could speak less than 5 words even though they were over 3 years old. Abnormal sensory processing, such as biting, touching, or smelling objects was common in these children. Three patients (Case 19/21/25) were overweight, and ptyalism still existed owing to hypotonia. Case 21 also had strabismus. It was specifically observed that Cases 21/25/26 with *MYT1L* variants had apparent stereotypic hand movements. Analysis of characteristics in these patients can better screen and classify them in the clinic.

#### 3.2.2. Renewed Case: Genetic Diagnosis Should Also Focus on Clinical Manifestations

Case 6 was a boy with compound heterozygous variants of *SLC26A4* (Figure 3B). The patient failed postnatal hearing screening and was inaudible to high-pitched sounds. He was diagnosed with bilateral profound sensorineural hearing loss and enlarged vestibular aqueduct syndrome soon afterwards. Right cochlear implants were performed when the child was 1 year old. He has been undertaking language training ever since. The patient was referred to our clinic at 4 years old because he behaved abnormally and undisciplined in kindergarten. His mother reported her child had a short attention span and hyperactivity. The Wechsler Preschool and Primary Scale of Intelligence (WPPSI) [22] showed that the child had developmental delay both in verbal and nonverbal intelligence (his IQS of Verbal Scale was 48, and IQS of Performance Scales was 62, producing a Full Scale IQ of 50). Compound heterozygous variants of *SLC26A4* (c.1226G>A(p. R409H), c.2168A>G(p. H723R)) were detected in this patient by CTPS. The parents were validated by Sanger sequence. The results showed that his missense variants were both inherited by his parents (paternal: c.1226G>A (p. R409H) in exon 10, maternal: c.2168A>G(p. H723R) in exon 19). The patient had poor social communication and was suspected autistic, however, no reports have shown the correlations between the *SLC26A4* gene and autism. Neither did he had interventions related to ASD, nor regularly followed up at the clinic. Atomoxetine hydrochloride was taken irregularly until school age. He had a re-examination at 7 years old. Maintaining Back-and-forth conversation was hard for him. He displayed poor social reciprocity and exhibited repetitive patterns of behavior (running back and forth repeatedly and watching traffic lights consistently) and sensory perception problems such as counting numbers and biting or smelling objects. His ADOS score was above the cutoff (social affect 5, repetitive score 3, severity 4). He was eventually diagnosed with ASD. The WES of the core family and CMA also performed according to his parents’ requirements but did not reveal any other causative variants.

## 4. Discussion

### 4.1. Clinical Benefits and Limits of CTPS

Genetics have a large contribution to ASD. Identifying a genetic etiology improving accuracy of counseling for patients and their families. Information about prognosis and recurrence risk is the most important benefit of genetic testing. The benefits also include preventing co-occurring medical conditions and avoiding unnecessary tests and harmful treatments [8]. It is necessary to advise patients and their families to undergo genetic evaluation.

Both rare inherited and de novo variations are relevant to ASD during early neurodevelopment. Kim et al. [23] provided evidence that rare inherited variations have a functional relationship with ASD in the developing brain. Sanders et al. [24] detected 2591 families from SSC and revealed that de novo variants were strongly associated with ASD, leading to a total of 6 risk loci (1q21.1, 3q29, 7q11.23, 16p11.2, 15q11.2-13, 22q11.2), and 65 ASD risk genes (additional 2 loci: *NRXN1* (2p16.3), *SHANK3*(22q13.3) were included in the list of risk genes). The CNVs detected in our study overlapped with their observation in 3q29, 16p11.2, 15q11.2-13, 22q11.2, and 22q13.3. These CNVs had a high risk of developmental delay [25]. Satterstrom et al.‘s work [26] undertook the largest exome sequencing, identified 102 ASD risk genes, most of which had effects on the regulation of gene expression or neuronal communication. Of these genes, 19 of them overlapped with our study (*ARID1B*, *ASXL3*, *BCL11A*, *CHD8*, *CTNNB1*, *DEAF1*, *FOXP1*, *FOXP2*, *KDM6B*, *MED13L*, *PPPP2R5D*, *PTEN*, *SCN2A*, *SETD5*, *SHANK2*, *SHANK3*, *SKI*, *STXBP1*, *TCF20*), whereas they had not analyzed de novo mutations on chromosome X. For high-confidence risk genes reported in Choi’s study [27], we had 16 genes overlapped (*ASXL3*, *BCL11A*, *CHD8*, *CTNNB1*, *DDX3X*, *DEAF1*, *FOXP1*, *FOXP2*, *KDM6B*, *MYT1L*, *PTEN*, *SCN2A*, *SETD5*, *SHANK3*, *SKI*, *TCF20*). It’s worth noting that the contribution of de novo non-coding variants may not be as high as that of coding regions [28].

For now, CMA and Fragile X testing are recommended as the first tier for ASD patients. CMA tests abnormalities of chromosomal structure and duplications or deletions in chromosomal regions. The diagnostic yield of CNV is 5.4–14% (median 9%) in ASD patients [29,30]. When CMA does not find an etiology, the next step recommended for etiologic evaluation of ASD is WES. WES identified SNVs that have been confirmed as ASD risk genes. The reported diagnostic yield is 8–25.8% [29,31]. This process takes much expenditure and is a waste of time. CTPS, comprising 2742 genes, is able to analyze SNVs and CNVs simultaneously to provide an etiological diagnosis for children and patients. The potential of CTPS is that it can discover variants effectively and costs less. Reducing the cost of time and price is much needed in developing countries, especially in China. Families will easily accept and be willing to undergo genetic testing and finally benefit from it. As research progresses, genetic testing may contribute to identifying effective interventions related to specific etiologies.

Based on the results of CTPS in this study, we had an overall diagnostic yield of 19.16% for 569 ASD patients, including both SNVs and CNVs. A meta-analysis in 2020 identified 14 ASD studies across 1530 patients using targeted gene panel sequencing or WES, and the diagnostic yield was 17.1% (95% CI, 11–25%) [32]. Rossi et al. [31] recruited 163 ASD/autistic patients, all of whom had additional clinical features such as intellectual disabilities (ID)/developmental delays (DD) (92.6%) and epilepsy/seizures (38.7%). They found that the diagnostic rate of their ASD cohort was 25.8% using WES (42 of 163). Aspromonte et al. [33] developed a next-generation sequencing gene panel of 74 selected genes. They analyzed 150 individuals with ID and/or ASD, and a confident diagnosis was reached in 41 patients (27%). In our study, the diagnostic yield was slightly lower than reported previously. The possible reason is that it was a single-center analysis, which may have selection bias in the sample. The children were referred to our department with fewer comorbidities, such as epilepsy or multiple other malformations. CTPS detected selected genes, not including the whole exome/genome. Additionally, a small number of CNVs were not covered in the panel [16]. The CTPS was designed for children with syndrome and developmental abnormalities, not a targeted panel specialized for ASD children. However, it had comprehensive candidate genes other than ASD genes to avoid missing suspected disease-causing genes. The restriction could explain the lower rate of our study.

Clinicians should focus more on the clinical features of children and find more valuable clues to the cause of the disease. CTPS offers a better choice for patients and clinicians as an effective molecular diagnostic tool.

### 4.2. Implications for Representative Cases from CTPS

Myelin transcription factor 1-like (*MYT1L*), mapping to a region of human chromosome 2, encodes the MYT1L protein, which contains 6 zinc fingers (an N-terminal zinc finger, 2 tandem central zinc fingers, and 3 C-terminal zinc fingers) [34]. The *MYT1L* gene is co-expressed with other ASD-related genes, including *NRXN1*, *TCF4*, and *BCL11A* in the prefrontal cortex during mid-fetal development, higher in prenatal expression [35], and transcription may persist in the brains of children. It plays a crucial role in neurogenesis, helps neural stem cells transform into neurons, and is expressed in oligodendrocyte linage cells during myelination and remyelination [36]. We found 4 children with de novo *MYT1L* variants. Coursimault et al. [37] reported 40 patients with *MYT1L*-associated neurodevelopmental disorder and reviewed 22 patients in published data. In their reports, many patients with *MYT1L* variants had language delay (95%), ID (70%), ASD (43%), motor delay (78%), and hypotonia (47%). Consistent with the previous literature, all of our 4 patients with ASD had developmental delays, overweight/obesity were common in these patients. Table 3 shows the clinical symptoms of our patients and public literature. All of our patients had abnormal sensory processing, whereas few studies mentioned it. Notably, 3 of our patients had stereotypic hand movements, which were not detailed in Juliette et al.’s work. De Rocker et al. [38] showed 3 patients with rigid hand movements, which was also manifested in our patients. These children had special hand phenotypes (typical repetitive, purposeless hand movements such as rubbing, tapping, wringing, or clapping) without rapid regression of acquired skills and stagnation. Therefore, a detailed history is particularly important. Our patients had obvious sensory abnormalities and stereotypic hand movements, which may contribute to the phenotype of *MYT1L*. 

Solute carrier family 26 member 4 (*SLC26A4*), encoding pendrin (an anion transporter) [44], causes Pendred syndrome and an enlarged vestibular aqueduct. It is expressed mostly in the thyroid and is also expressed at some levels in the prostate, kidney, urinary bladder, and brain [45]. Pedred Syndrome is a common disorder with hereditary hearing loss, abnormal development of the cochlea, and diffuse thyroid enlargement. The patient in our study had *SLC26A4* heterozygous variants. Due to the lack of literature connecting *SLC26A4* with autism, CMA and WES were performed. The patient had typical autistic features but found no cause other than *SLC26A4* variants. He had a complex phenotype beyond Pendred syndrome presentation, hence, our study reported an ASD child with *SLC26A4* variants. It is hard for us to explain his sensory perception abnormalities and repetitive behaviors with hearing problems. *SLC26A4* variants may cause his autistic phenotype, while early social competence may be overlooked by deafness. Nevertheless, further studies and case reports are required to show the relationships between the *SLC26A4* gene and phenotype with ASD. After all, some of the variants of uncertain significance may be determined as pathogenic in the future [8]. It is hoped that more animal research will be conducted to examine the effect of *scl26a4* on autism and on the gene-phenotype associations in neurodevelopment.

## 5. Conclusions

In summary, CTPS is an effective tool for genetic testing of ASD patients, a valuable strategy suitable for patients with neurodevelopmental disorders in China. It is important to identify the key clinical features of patients diagnosed with ASD that allow more accurate genetic diagnosis and harbor relevant genetic variants. It is expected that genetic research will allow the development of better treatments and interventions for ASD children within the next decade, thus guiding further family planning and social support.

## Figures and Tables

**Figure 1 genes-13-01010-f001:**
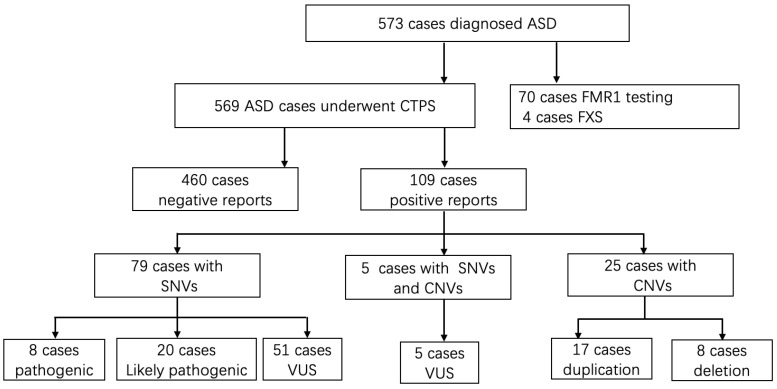
The flow of genetic evaluation of 573 cases with autism spectrum disorders in genetic clinics. ASD, autism spectrum disorder; CTPS: clinical targeted panel sequencing; FXS: Fragile X syndrome; SNVs: single nucleotide variations; CNVs, copy number variations; VUS, variant of unknown significance.

**Figure 2 genes-13-01010-f002:**
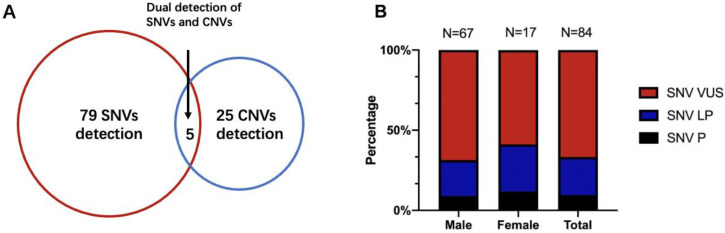
(**A**) The detection yield of SNV and CNV of CTPS in 569 patients. (**B**) The percentage of SNVs diagnostic variations between males and females. SNVs, single nucleotide variations; CNVs, copy number variations; P, pathogenic; LP, likely pathogenic; VUS, variant of unknown significance.

**Figure 3 genes-13-01010-f003:**
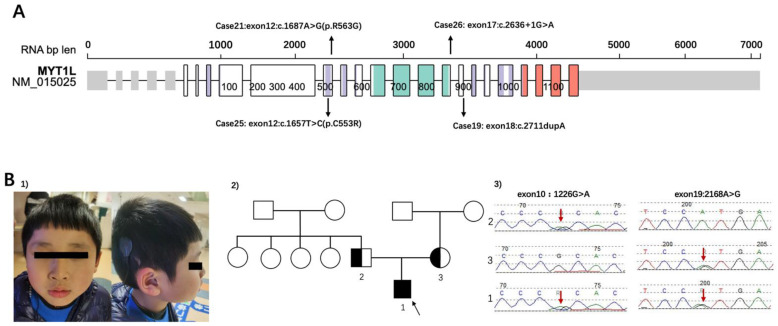
(**A**) The locations of variants of patients in the study. Representation based on Protein Paint (Wisteria color: zf-C2HC domain, Turquoise color: MYT1 domain, Orange color: Smc domain) (https://proteinpaint.stjude.org/, accessed on 1 March 2022). The cases correspond with Table 1. (**B**) (1) The dysmorphic features of case 6 (shown for 1). (2) The Pedigree of family 1. 2 was the father of the patient; 3 was the mother of the patient. The black arrow showed the patient of case 6. (3) *SLC26A4* sequence results of patient and his parents. Heterozygous variants of *SLC26A4* were identified in the proband (red arrows in patient 1). Both his parents were in a heterozygous state for the variant (paternal: red arrow of 2 for c.1226G>A (p. R409H) in exon 10, maternal: red arrow of 3 for c.2168A>G(p. H723R) in exon 19).

**Table 1 genes-13-01010-t001:** SNVs identified from ASD patients from CTPS.

Case	Sex	Classification	Gene	Position	Mutation	Zygosity	Inherited/ de Novo	Inheritance Pattern	Population Frequency
Pathogenic									
1	M	P	*SHANK3*	chr22:51159988	NM_033517:exon21:c.3727C>T(p.Q1243X)	Het	AD	De novo	NA
2	M	P	*TRIP12*	chr2:230744756	NM_004238:exon2:c.40C>T(p.R14X)	Het	AD	De novo	NA
3	M	P	*NF1*	chr17:29552152	NM_000267:exon17:c.1885G>A(p.G629R)	Het	AD	De novo	3.98 × 10^−6^
4	M	P	*RAB39B/* *NDST1/* *NDST1*	chrX:154493531/ chr5:149907553/ chr5:149929268	NM_171998:exon1:c.43G>C(p.G15R)/ NM_001543:exon3:c.701C>T(p.T234I)/ NM_001543:exon13:c.2345G>T(p.R782L)	Hemi/ Het/ Het	X-linked/ AR/ AR	NA	NA/ NA/ 4.24 × 10^-5^
5	M	P	*TCF20*	chr22:42608728	NM_005650:exon1:c.2582_2583del	Het	AD	De novo	NA
6	M	P	*SLC26A4/* *SLC26A4*	chr7:107330645/ chr7:107350577	NM_000441:exon10:c.1226G>A(p.R409H)/NM_0004411:exon19:c.2168A>G(p.H723R)	Het/ Het	AR/ AR	Paternal/ Maternal	9.57 × 10^−5^/ 1.13 × 10^−4^
7	F	P	*SHANK3*	chr22:51158733	NM_033517:exon21:c.2475dupC	Het	AD	De novo	NA
8	F	P	*MECP2*	chrX:153296399	NM_004992:exon4:c.880C>T(p.R294X)	Het	XLD/XLR	NA	NA/
Likely pathogenic									
9	M	LP	*BRAF*	chr7:140453146	NM_004333:exon15:c.1789C>G(p.L597V)	Het	AD	NA	NA
10	M	LP	*DIP2B*	chr12:51128855	NM_173602:exon34:c.4044-1G>A	Het	AD	NA	NA
11	M	LP	*KDM6B*	chr17:7752387_7752400	NM_001080424:exon11:c.2781_2794del14;p.(Val928Hisfs*2)	Het	AD	NA	NA
12	M	LP	*SCN2A*	chr2:166201068	NM_021007:exon16:c.2566C>T(p.R856X)	Het	AD	NA	NA
13	M	LP	*STXBP1*	chr9:130432185	NM_003165:exon11:c.913dupC	Het	AD/AR	Maternal	NA
14	M	LP	*CHD8*	chr14:21871817	NM_001170629:exon16:c.3312delT	Het	AD	De novo	NA
15	M	LP	*DEAF1*	chr11:687941	NM_021008:exon4:c.634G>A(p.G212S)	Het	AD/AR	Maternal	NA
16	M	LP	*AP1S2*	chrX:15870644	NM_003916:exon2:c.4C>T(p.Q2X)	Hemi	XLR	Maternal	NA
17	M	LP	*EHMT1/* *FGFR3*	chr9:140678599/ chr4:1807288	NM_001145527:exon16:c.2424G>A(p.W808X)/ NM_000142:exon12:c.1537G>A(p.D513N)	Het/ Het	AD/ AD/AR	Maternal/ Maternal	NA/ 4.61 × 10^−5^
18	M	LP	*ABCD1*	chrX:152991143	NM_000033:exon1:c.422C>T(p.A141V)	Hemi	XLR	NA	NA
19	M	LP	*MYT1L*	chr2:1890310	NM_015025:exon18:c.2711dupA	Het	AD	De novo	NA
20	M	LP	*DYRK1A*	chr21:38865398	NM_001396:exon7:c.1031_1037del	Het	AD	NA	NA
21	M	LP	*MYT1L*	chr2:1915808	NM_015025:exon12:c.1687A>G(p.R563G)	Het	AD	De novo	NA
22	M	LP	*BCL11A*	chr2:60689252	NM_022893:exon4:c.793_794delCT	Het	AD	De novo	NA
23	M	LP	*ARID2*	chr12:46244889	NM_152641:exon15:c.2983C>T(p.Q995X)	Het	AD	De novo	NA
24*	F	LP	*MECP2*	chrX:153296399	NM_004992:exon4:c.880C>T(p.R294X)	Het	XLD/XLR	De novo	1.64 × 10^−5^
25	F	LP	*MYT1L*	chr2:1915838	NM_015025:exon12:c.1657T>C(p.C553R)	Het	AD	De novo	NA
26	F	LP	*MYT1L*	chr2:1891259	NM_015025:exon17:c.2636+1G>A	Het	AD	De novo	NA
27	F	LP	*SHANK3*	chr22:51159684	NM_033517:exon21:c.3424_3425delCT	Het	AD	NA	NA
28	F	LP	*SLC2A1*	chr1:43395422	NM_006516:exon6:c.709G>A(p.V237M)	Het	AD/AR	NA	7.95 × 10^-6^
Variant of unknown significance (VUS)									
29	M	VUS	*TRIO*	chr5:14508475	NM_007118:exon57:c.9238C>T(p.R3080X)	Het	AD	Paternal	NA
30	M	VUS	*TRIO*	chr5:14474205	NM_007118:exon40:c.6082G>A(p.D2028N)	Het	AD	NA	3.98 × 10^−6^
31	M	VUS	*DCLRE1C/* *DCLRE1C/* *KANK1/* *KIF1A*	chr10:14976777/ chr10:14981850/ chr9:712617/ chr2:241702608	NM_001033855:exon7:c.465-3C>T/ NM_001033855:exon4:c.265A>G(p.T89A)/ NM_015158:exon3:c.1852del/ NM_004321:exon19:c.1897G>A(p.D633N)	Het/ Het/ Het/ Het	AR/ AR/ AD/ AD/AR	NA	NA/ 2.85 × 10^−5^/ NA/ NA
32	M	VUS	*SCN8A*	chr12:52080904	NM_014191:exon5:c.515A>T(p.E172V)	Het	AD	Maternal	NA
33	M	VUS	*KIF7*	chr15:90172648	NM_198525:exon17:c.3472_3474delAAG	Hom	AR	NA	8.13 × 10^−6^
34	M	VUS	*SMARCA4*	chr19:11105513	NM_001128849:exon9:c.1429A>G(p.N477D)	Het	AD	De novo	NA
35	M	VUS	*MED13L*	chr12:116418709	NM_015335:exon23:c.5210A>G(p.K1737R)	Het	AD	NA	NA
36	M	VUS	*SKI/* *WDR81/* *WDR81*	chr1:2161179/ chr17:1633670/ chr17:1636940	NM_003036:exon1:c.969+5G>C/ NM_001163809:exon2:c.3668-4G>A/ NM_001163809:exon7:c.4609G>A(p.G1537S)	Het/ Het/ Het	AD/ AR/ AR	De novo/ Maternal/ Paternal	NA/ 4.34 × 10^−5^/ 4.14 × 10^−5^
37	M	VUS	*MAOA*	chrX:43552577	NM_000240:exon3:c.208G>A(p.V70M)	Hemi	XLR	Maternal	NA
38	M	VUS	*NCAPD3/* *NCAPD3*	chr11:134022951/ chr11:134051019	NM_015261:exon35:c.4389-4C>G/ NM_015261:exon20:c.2512A>G(p.I838V)	Het/ Het	AR/ AR	Paternal/ Maternal	NA/ 1.59 × 10^−5^
39	M	VUS	*IQSEC2*	chrX:53285098	NM_001111125:exon3:c.883C>T(p.R295W)	Hemi	XLD	NA	NA
40	M	VUS	*DYRK1A*	chr21:38884272	NM_001396:exon11:c.1730T>A(p.V577D)	Het	AD	NA	NA
41	M	VUS	*PIGV/* *PIGV*	chr1:27121024/ chr1:27121133	NM_017837:exon3:c.499G>A(p.G167S)/ NM_017837:exon3:c.608G>T(p.R203L)	Het/ Het	AR/ AR	NA	3.54 × 10^−5^/ NA
42	M	VUS	*FOXP2*	chr7:114174736	NM_014491:exon3:c.233G>C(p.S78T)	Het	AD	NA	NA
43	M	VUS	*HCFC1*	chrX:153219744	NM_005334:exon17:c.4106T>C(p.M1369T)	Hemi	XLR	Maternal	NA
44	M	VUS	*DYRK1A*	chr21:38850489	NM_001396:exon3:c.214C>G(p.P72A)	Het	AD	NA	NA
45	M	VUS	*PTEN*	chr10:89717672	NM_000314:exon7:c.697C>T(p.R233X)	Het	AD/AR	NA	NA
46	M	VUS	*SLITRK1*	chr13:84455351	NM_052910:exon1:c.292G>A(p.V98I)	Het	AD	NA	NA
47	M	VUS	*PLA2G6/* *PLA2G6*	chr22:38516880/ chr22:38528924	NM_003560:exon12:c.1628G>A(p.R543H)/ NM_003560:exon7:c.991G>T(p.D331Y)	Het/ Het	AR/ AR	NA	2.83 × 10^−5^/ 3.57 × 10^−5^
48	M	VUS	*PHIP*	chr6:79727249	NM_017934:exon11:c.1046T>A(p.F349Y)	Het	AD	Paternal	NA
49	M	VUS	*CTNNB1/* *HCFC1/* *SOS1*	chr3:41266085/ chrX:153217162/ chr2:39216457	NM_001904:exon3:c.82C>G(p.Q28E)/ NM_005334:exon21:c.5261-4C>T/ NM_005633:exon21:c.3347-2A>G	Het/ Hemi/ Het	AD/ XLR/ AD	NA	NA/ 1.85 × 10^−5^/ NA
50	M	VUS	*GRIA3*	chrX:122318409	NM_000828:exon1:c.22G>A(p.G8R)	Hemi	XLR	NA	NA
51	M	VUS	*COL4A3BP*	chr5:74712810	NM_001130105:exon8:c.1112G>A(p.G371E)	Het	AD	NA	NA
52	M	VUS	*DPP6*	chr7:154585802	NM_001936:exon11:c.964A>G(P.T322A)	Het	AD	NA	NA
53	M	VUS	*PHIP/* *ACVR1*	chr6:79672916/ chr2:158626989	NM_017934:exon30:c.3433A>G(p.R1145G)/ NM_001105:exon7:c.681G>A(p.W227X)	Het/ Het	AD/ AD	NA	NA/ NA
54	M	VUS	*SETBP1*	chr18:42531731	NM_015559:exon4:c.2426A>G(p.Q809R)	Het	AD	Paternal	NA
55	M	VUS	*FOXP1*	chr3:71015071	NM_032682:exon20:c.1859G>A(p.S620N)	Het	AD	NA	NA
56	M	VUS	*PHIP*	chr6:79665392	NM_017934:exon33:c.3790A>G(p.T1264A)	Het	AD	Paternal	NA
57	M	VUS	*DDX3X/* *DLG3*	chrX:41196685/ chrX:69719742	NM_001193416:exon2:c.70T>G(p.S24A)/ NM_021120:exon16:c.1988G>A(p.R663Q)	Hemi/ Hemi	XL/ XL	Maternal/ Maternal	NA/ NA
58	M	VUS	*GNAI3/* *USP27X*	chr1:110121866/ chrX:49645815	NM_006496:exon4:c.344A>G(p.E115G)/ NM_001145073:exon1:c.905T>C(p.L302S)	Het/ Hemi	AD/ XL	NA	NA/ NA
59	M	VUS	*TMLHE*	chrX:154743783	NM_018196:exon4:c.502C>T(p.Q168X)	Hemi	XLR	NA	NA
60	M	VUS	*BRWD3*	chrX:80064545	NM_153252:exon3:c.91-4T>C	Hemi	XLR	NA	NA
61	M	VUS	*USP9X*	chrX:41029747	NM_001039590:exon20:c.2902A>C(p.I968L)	Hemi	dominant/XLR	Maternal	NA
62	M	VUS	*FOXP1*	chr3:71037180	NM_032682:exon14:c.1111G>A(p.V371M)	Het	AD	NA	NA
63	M	VUS	*DIP2B*	chr12:51074491	NM_173602:exon9:c.1151C>T(p.T384I)	Het	AD	NA	2.12 × 10^−5^
64	M	VUS	*L1CAM*	chrX:153133875	NM_000425:exon13:c.1585G>A(p.E529K)	Hemi	XLR	NA	NA
65	M	VUS	*SHANK3*	chr22:51169394	NM_033517:exon22:c.4850C>T(p.P1617L)	Het	AD	NA	NA
66	M	VUS	*DIP2B*	chr12:51068356	NM_173602:exon6:c.740T>C(p.I247T)	Het	AD	NA	3.18 × 10^−5^
67	M	VUS	*GRIA3*	chrX:122551611	NM_000828:exon11:c.1859G>C(p.G620A)	Hemi	XLR	NA	NA
68	M	VUS	*AFF2/* *TRIO*	chrX:147743835/ chr5:14387875	NM_002025:exon3:c.587T>C(p.F196S)/ NM_007118:exon23:c.3800G>A(p.S1267N)	Hemi/ Het	XLR/AD	NA	NA/ 3.98 × 10^−6^
69	M	VUS	*FOXP2*	chr7:114303569	NM_014491:exon15:c.1834T>A(p.L612M)	Het	AD	NA	NA
70	M	VUS	*ARID1B/* *CHD7*	chr6:157521844/ chr8:61654295	NM_020732:exon18:c.4116C>A(p.Y1372X)/ NM_017780:exon2:c.304C>T(p.H102Y)	Het/ Het	AD/ AD	De novo/ Paternal	NA/ NA
71 *	M	VUS	*CTNNB1/* *KLHL15*	chr3:41279547/ chrX:24006703	NM_001904:exon14:c.2117C>A(p.P706H)/ NM_030624:exon4:c.1150G>A(p.V384I)	Het/ Hemi	AD/ XLR	NA	NA/ 5.48 × 10^−6^
72 *	M	VUS	*RPS6KA3*	chrX:20284690	NM_004586:exon1:c.61A>G(p.S21G)	Hemi	XLD	NA	NA
73 *	M	VUS	*KIF1A/* *ZC4H2*	chr2:241700653/ chrX:64137775	NM_004321:exon22:c.2231A>G(p.K744R)/ NM_018684:exon5:c.563C>T(p.A188V)	Het/ Hemi	AD/AR/ XLR	De novo/ Maternal	NA/ NA
74 *	M	VUS	*FBN2/* *SPTAN1*	chr5:127625581/ chr9:131389713	NM_001999:exon51:c.6503delC/ NM_001130483:exon50:c.6625G>A(p.D2209N)	Het/ Het	AD/ AD	NA	NA/ 6.34 × 10^−6^
75	F	VUS	*SHANK3*	chr22:51142293	NM_033517:exon13:c.1618C>T(p.R540W)	Het	AD	NA	NA
76	F	VUS	*GPR98/* *GPR98*	chr5:89954046/ chr5:90074281	NM_032119:exon21:c.4703G>A(p.S1568N)/ NM_032119:exon63:c.12704A>G(p.Y4235C)	Het/ Het	AD/AR/Digenic/ AD/AR/Digenic	NA	5.46 × 10^−5^/ 1.61 × 10^−4^
77	F	VUS	*PPP2R5D*	chr6:42974971	NM_006245:exon5:c.560C>T(p.S187L)	Het	AD	NA	NA
78	F	VUS	*MECP2*	chrX:153296071	NM_004992:exon4:c.1158_1201del	Het	XLD/XLR	NA	NA
79	F	VUS	*SETBP1*	chr18:42529856	NM_015559:exon4:c.551G>T(p.R184M)	Het	AD	NA	NA
80	F	VUS	*SETD5*	chr3:9512347	NM_001080517:exon19:c.2929T>A(p.F977I)	Het	AD	Paternal	NA
81	F	VUS	*HUWE1/* *LRP2/* *LRP2*	chrX:53574690/ chr2:170030607/ chr2:170081950	NM_031407:exon68:c.10580T>C(p.V3527A)/ NM_004525:exon56:c.10836G>T(p.Q3612H)/ NM_004525:exon33:c.5406_5407del	Het/ Het/ Het	XL/ AR/ AR	NA	NA/ NA NA
82	F	VUS	*ERCC2/* *ERCC2/* *ASXL3*	chr19:45868096-45868099/ chr19:45856520/ chr18:31322948	NM_000400:exon7:c.591_594del/ NM_000400:exon18:c.1738G>A(p.A508T)/ NM_030632:exon12:c.3136G>A(p.G1046R)	Het/ Het/ Het	AR/ AR/ AD	Paternal/Maternal/Maternal	1.20 × 10^−5^/ 1.59 × 10^−5^/ 1.61 × 10^−5^
83	F	VUS	*FOXP1*	chr3:71247424	NM_032682:exon6:c.109T>C(p.S37P)	Het	AD	NA	NA
84 *	F	VUS	*DCX*	chrX:110574270	NM_178153:exon5:c.809-1G>C	Het	XL	NA	NA

SNV, single nucleotide variations; M, male; F, female; Hom, homozygous; Het, heterozygous; Hemi, hemizygous; AD, autosomal dominant; AR, autosomal recessive; XL, X-linked; XLD, X-linked dominant; XLR, X-linked recessive; P, pathogenic; LP, likely pathogenic; VUS, variant of unknown significance. *, patients with dual SNV and CNV.

**Table 2 genes-13-01010-t002:** CNVs identified from ASD patients from CTPS.

Case	Sex	Band	Chr	Start(hg19)	Stop(hg19)	Size(kb)	Deletion/Duplication
1 *	M	15q13.2–15q13.3	chr15	30,653,442	32,464,722	1,811,280	deletion
2 *	M	20p12.1–20p13	chr20	740,723	13,799,067	13,058,344	duplication
3 *	M	16p11.2	chr16	29,802,039	30,200,397	398,358	deletion
4 *	M	Xq28	chrX	153,576,898	153,780,404	203,506	duplication
5 *	F	15q11.2–15q13.1	chr15	23,043,276	28,327,041	5,283,765	duplication
6	M	7p13–7p14.1	chr7	41,724,711	44,748,665	3,023,954	duplication
7	M	15q13.3	chr15	32,064,983	32,443,563	378,580	duplication
8	M	15q11.2–15q13.1	chr15	23,043,276	28,327,041	5,283,765	duplication
9	M	2q24.3–2q25.1	chr2	9,628,275	16,087,129	6,458,854	duplication
10	M	15q11.2–15q13.1	chr15	23,043,276	28,327,041	5,283,765	duplication
11	M	3q29	chr3	196,195,653	197,024,106	828,453	deletion
12	M	1p21.2–1p21.3	chr1	97,543,298	100,715,390	3,172,092	duplication
13	M	15q13.2–15q13.3	chr15	30,659,620	32,464,722	1,805,102	duplication
14	M	7q36.1–7q36.3	chr7	150,642,048	157,210,133	6,568,085	deletion
15	M	Xp21.1	chrX	32,235,032	32,235,180	148	deletion
16	M	19p13.2–q13.3	chr19	43,370,615	43,530,621	160,006	deletion
17	M	10q22.3–10q23.2	chr10	81,697,495	88,854,623	7,157,128	duplication
18	M	4q35.1–q 35.2	chr4	186,421,813	190,873,442	4,451,629	deletion
19	M	6q16.1–6q16.3	chr6	97,337,188	105,307,794	7,970,606	deletion
20	M	15q11.2–15q13.1	chr15	23,043,276	28,327,041	5,283,765	duplication
21	M	3q29	chr3	195,776,154	197,300,194	1,524,040	deletion
22	M	1p34.3	chr1	36,974,539	38,129,928	1,155,389	duplication
23	F	22q13.31–22q13.33	chr22	45,680,862	51,171,726	5,490,864	deletion
24	F	14q21.1	chr14	39,559,493	39,665,452	105,959	duplication
25	F	2q37.3	chr2	240,016,194	242,708,226	2,692,032	deletion
26	F	17p11.2	chr17	16,664,738	20,370,848	3,706,110	duplication
27	F	10q22.3–q23.2	chr10	81,697,495	88,854,623	7,157,128	duplication
28	F	22q11.21	chr22	18,900,293	21,245,506	2,345,213	duplication
29	F	2q37.12q37.3	chr2	234,408,524	242,844,702	8,436,178	deletion
30	F	p21.1	chrX	32,305,645	32,632,570	326,925	duplication

*: patients with dual SNV and CNV (case 1, 2, 3, 4, and 5 were case 71, 72, 73, 74, and 84 in Table 1); CNV, copy number variations; Chr, chromosome; M, male; F, female.

**Table 3 genes-13-01010-t003:** Clinical symptoms of *MYT1L* patients and published literature.

	Autism Spectrum Disorder	Language Delay	Motor Delay	Developmental Disorder/ Intellectual Disability	Stereotypic Hand Movements	Abnormal Sensory Processing	Hypotonia	Overweight/ Obesity
Our study	100% (4/4)	100% (4/4)	100% (4/4)	100% (4/4)	75% (3/4)	100% (4/4)	75% (3/4)	75% (3/4)
Coursimault et al.‘s [ [37]	43% (17/40)	95% (38/40)	78% (31/40)	70% (21/30)	-	-	47% (18/38)	58% (23/40)
De Rocker et al.’s [38]	32% (7/22)	100% (22/22)	-	100% (22/22)	14% (3/22)	-	-	74% (14/19)
Windheuser et al.’s [39]	22% (2/9)	-	87% (7/8)	100% (8/8)	-	Mentioned in 1 patient	78% (7/9)	33% (3/9)
Blanchet et al.’s [40]	44% (4/9)	100% (9/9)	100% (8/8)	-	-	-	Mentioned in 2 patients	66.7% (6/9)
Carvalho et al.’s [41]	0	100% (1/1)	-	100% (1/1)	-	-	-	100% (1/1)
Loid et al.’s [42]	0	100% (1/1)	100% (1/1)	100% (1/1)	-	-	-	100% (1/1)
Al Tuwaijri et al.’s [43]	100% (1/1)	100% (1/1)	100% (1/1)	100% (1/1)	-	-	100% (1/1)	100% (1/1)

## Data Availability

The data presented in this study are available on request from the corresponding author. The confidential patient data will not be shared.

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
