# Peer review of "Clinical Targeted Panel Sequencing Analysis in Clinical Evaluation of Children with Autism Spectrum Disorder in China"

_genes, 2022, doi:10.3390/genes13061010_

Round 1

Reviewer 1 Report

In this manuscript, Hu et al. reported their study using clinical exome sequencing in children with ASD. They analyzed 2742 genes in 573 patients, and identified SNV and CNV with the sequencing data. Especially recurrently seen variants in genes including SHANK3, MYT1L, SLC26A4, etc. Their findings may allow the development of better treatments and interventions for children with ASD.

Overall, this study is well designed, and performed appropriately. The findings of this manuscript could potentially benefit the area of ASD research, and the patient data are also useful for the field. Several major issues should be addressed.

Major issues:

  1. The authors defined their assay as “Clinical Exome Sequencing (CES)”. However, this is misleading because in the general understanding, exome sequencing always means targeting almost all (usually ~ 20000 genes) genes’ exons. This study used Agilent ClearSeq Inherited Disease panel kit to target 2742 genes. Clinical targeted panel sequencing would be the better definition of this.

  2. The authors used the concept of “overall positive rate” for SNV and CNV detection. However, the definition of this overall positive rate is not clearly described. Moreover, in section 2.2, the authors did not tell the detail of the filtrations and annotation procedures and criteria for the analysis; and the authors did not cite proper references that described this information. Missing descriptions of the methods and a clear definition of the “positive rate” could make it difficult for readers to understand their results.

  3. Again, the materials and methods are not clearly described. I cannot even find what tissue they used for DNA extraction, or sequencing strategy (PE? read length? data quality and coverage? etc). In section 2.2, the authors should cite the references for the software used in their study, and provide a clear description of the software version and parameters.

  4. Table 2, for the CNV reported in this work, the authors should include the information about what genes are impacted, especially those with important functions.

Minor issues:

  1. Line 95, the authors should include a description of the reason for doing the FMR1 test in those 70 patients.

  2. In table 1, some critical annotations should be included, like population frequency.

Reviewer 2 Report

The manuscript by Hu et al. reports clinical exome sequencing of 573 patients with autism spectrum disorder (ASD). They identified SNV and CNVs possibly causing ASD in patients: 13.89% (79/569) and 4.40% (25/569) of cases have SNVs only and 20 CNVs. From systematic review on identified variants and affected genes, they found SHANK3 and MYT1L gene frequently seen in their patients. MYT1L is very interesting to the field. I think the study was thoroughly done but required some points to be clarified.

  1. It seems that the exome analysis was done for only patients. The authors "causal" filtered variants but there is no method described how it's been done. They found rare or de novo variants and examine affected genes with possible causation. Thus, the method (e.g. allele frequency filtering, pathogenicity prediction, etc.) to prioritize causal variants has to be filled. This study might help some guide to describe the analytic framework (https://www.mdpi.com/2073-4425/12/1/75).

  1. I am wondering whether you see any overlap between the affected genes (Table 1) and the possible causal genes from the latest large-scale genomic studies? There are some studies to compare:

- 102 genes for de novo mutation from the largest exome: https://pubmed.ncbi.nlm.nih.gov/31981491/

- Or you can use Table 3 from this review: https://www.sciencedirect.com/science/article/pii/S0149763421002700

  1. I cannot find the list of identified CNVs including their location, annotation, and frequency in supplementary table. The authors should provide this information. Please see the format from Table S2 in this paper. https://pubmed.ncbi.nlm.nih.gov/26402605/

  1. For Table 2, how many CNVs identified from ASD patients from CES are overlapped with known ASD or developmental delay CNVs? You can compare with the study above (PMID 26402605) or the study that summarize CNV loci for developmental delay and ASD (https://pubmed.ncbi.nlm.nih.gov/28645357/).

  1. The author described MYT1L is expressing in brain. There is more specific dataset for developing human prefrontal cortex (PMID 32268104). MYT1L is highly expressed in fetal brain and shows strong co-expression pattern with other developmental genes. Table S2 from the study of PMID 32268104 has information on this gene and its expressing module - M4. This has to be discussed as well.

  1. Many key citations on ASD CNV or exome studies are missing. These recommend to be cited in the reference:

- Sanders 2015 Neuron, https://pubmed.ncbi.nlm.nih.gov/26402605/

- Kim 2020, Genes, https://www.mdpi.com/2073-4425/11/5/535

- Werling 2018 Nature Genetics, https://www.nature.com/articles/s41588-018-0107-y

- Lowther 2020, bioRxiv, https://www.biorxiv.org/content/10.1101/2020.08.12.248526v1.abstract

- Satterstrom 2021 Cell, https://pubmed.ncbi.nlm.nih.gov/31981491/

Round 2

Reviewer 1 Report

The authors addressed all my concerns and made good improvements to the manuscript. I agree it can be accepted now.

Reviewer 2 Report

I don't have any further concern for this revision.